# GERD after Bariatric Surgery. Can We Expect Endoscopic Findings?

**DOI:** 10.3390/medicina57050506

**Published:** 2021-05-17

**Authors:** Ramon Vilallonga, Sergi Sanchez-Cordero, Nicolas Umpiérrez Mayor, Alicia Molina, Arturo Cirera de Tudela, Elena Ruiz-Úcar, Manel Armengol Carrasco

**Affiliations:** 1Endocrine, Metabolic and Bariatric Unit, Department of General and Digestive Surgery, Center of Excellence for the EAC-BC, Vall d’Hebron University Hospital, Universitat Autònoma de Barcelona, Passeig de la Vall d’Hebron 119-129, 08035 Barcelona, Spain; vilallongapuy@gmail.com; 2ELSAN, Clinique Saint Michel, Centre Chirurgical de l’Obésité, 83100 Toulon, France; 3Department of General and Digestive Surgery, Igualada University Hospital, Av. Catalunya 11, Igualada, 08700 Barcelona, Spain; 4Department of General and Digestive Surgery, Vall d’Hebron University Hospital, Universitat Autònoma de Barcelona, Passeig de la Vall d’Hebron 119-129, 08035 Barcelona, Spain; nicolasumpierrezmayor@gmail.com (N.U.M.); a.cireradetudela@gmail.com (A.C.d.T.); marmenol@vhebron.net (M.A.C.); 5UCON, Obesity and Nutrition Surgery Unit, Corachan Clinic, Tres Torres, 7, 08017 Barcelona, Spain; consultesexternescmq@grupsagessa.cat; 6Endocrine, Metabolic and Bariatric Unit, Department of General and Digestive Surgery, Fuenlabrada University Hospital, Rey Juan Carlos University, Camino del Molino, 2, 28942 Madrid, Spain; eruizucar@gmail.com

**Keywords:** gastroesophageal reflux disease, endoscopy, bariatric surgery, sleeve gastrectomy, gastric bypass, hiatal hernia, obesity, GERD

## Abstract

*Background and Objectives**:* Bariatric surgery remains the gold standard treatment for morbidly obese patients. Roux-en-y gastric bypass and laparoscopic sleeve gastrectomy are the most frequently performed surgeries worldwide. Obesity has also been related to gastroesophageal reflux disease (GERD). The management of a preoperative diagnosis of GERD, with/without hiatal hernia before bariatric surgery, is mandatory. Endoscopy can show abnormal findings that might influence the final type of surgery. The aim of this article is to discuss and review the evidence related to the endoscopic findings after bariatric surgery. *Materials and Methods*: A systematic review of the literature has been conducted, including all recent articles related to endoscopic findings after bariatric surgery. Our review of the literature has included 140 articles, of which, after final review, only eight were included. The polled articles included discussion of the endoscopy findings after roux-en-y gastric bypass and laparoscopic sleeve gastrectomy. *Results:* We found that the specific care of bariatric patients might include an endoscopic diagnosis when GERD symptoms are present. *Conclusions:* Recent evidence has shown that endoscopic follow-up after laparoscopic sleeve gastrectomy could be advisable, due to the pathological findings in endoscopic procedures in asymptomatic patients.

## 1. Introduction

Bariatric surgery (BS) is the most effective treatment for morbid obesity and obtains the best long-term outcomes [1]. Bariatric surgery is also the only treatment option that achieves sustained weight loss and has a positive impact on related co-morbidities [2]. Gastroesophageal reflux disease (GERD) is recognized as a linked condition to obesity, especially morbid obesity [1,3,4]. This association between obesity and GERD is very well-known. In comparison to the general population, obese patients have 2–2.5 more chances of developing reflux symptoms [5,6]. More than 50% of obese patients develop GERD symptoms, and the condition is found in up to 70% of morbidly obese patients who seek bariatric surgery [7,8]. GERD is a common condition with a prevalence of 9% to 25% in Europe [9] and it is increasing in Eastern and Western countries [3]. Part of the problem with GERD is that sometimes we cannot find a correlation between self-reported reflux symptoms and their correlation with objectified reflux, meaning that a high percentage of patients with severe GERD symptoms do not have true pathological GERD on objective testing.

Laparoscopic sleeve gastrectomy (LSG) is currently the most performed surgery worldwide, followed by laparoscopic Roux-en-Y gastric bypass (LRYGB) [10,11]. Laparoscopic sleeve gastrectomy (LSG) seems to achieve equal weight loss as laparoscopic Roux-en-Y bypass (LRYGB), but there is still much debate about other aspects, including metabolic co-morbidity results, long term weight loss, and the quality of life after LSG, mainly concerning the association with gastroesophageal reflux [2]. Mid and long-term follow-up of GERD after bariatric surgery is controversial, however, data suggest that some procedures, such as LSG, could have more severe endoscopic findings compared to clinical symptoms [8]. There is no clear data regarding the correlation between GERD and endoscopic findings, however, some recent reports suggest that there is a need for endoscopic follow-up after LSG to avoid potential serious complications. The aim of this review article is to evaluate the relevant endoscopic findings in patients with GERD symptoms after LRYGB and LSG. Additionally, this review aims to elucidate the appearance of GERD in patients after LSG and LRYGB, and the endoscopic findings and correlations with clinical symptoms.

## 2. Materials and Methods

### 2.1. Search Strategy and Study Selection

An online search was conducted in PubMed/MEDLINE in February 2021 to identify articles reporting gastroesophageal reflux disease (GERD) and endoscopic findings after bariatric surgery. The eligible studies were selected following PRISMA guidelines. The searched keywords used were “gastroesophageal reflux”, “GERD”, “de novo GERD”, “endoscopy”, “Roux-en-Y gastric bypass”, or “following sleeve gastrectomy” or “following LSG”. The reference list of the selected studies was also manually checked to identify relevant papers. The conference abstracts considered as “gray literature”, were not selected. No publication date or language limit was considered for our search strategy.

Two authors searched and reviewed the full text of all clinical studies. The reference lists of selected studies were examined to obtain other relevant articles. The inclusion criteria of the review were the management or treatment of GERD after LSG or LRYGB. The most relevant, or comprehensive publications were finally included in the analysis to avoid duplicates, ambiguity, or papers reporting data from the same study population. 

A total of 140 records were identified by the initial search of the PubMed/MEDLINE databases. Of these, 125 papers were excluded after screening by title and abstract. Our initial analysis included a prescreen to identify the clearly irrelevant reports by title, abstract, and keywords of the publication. Relevant papers, including short series, review papers, and meta-analysis were kept for the further review process. Two other independent reviewers (AC. and SS.) then assessed the studies for relevance, inclusion, and methodological quality. The studies were classified as relevant (meeting all specified inclusion criteria); possibly relevant (meeting some but not all inclusion criteria); and rejected (not relevant to our review). Two reviewers (ER. and RV.) independently reviewed the full-text versions of all studies classified as relevant or possibly relevant. Any disagreements were resolved by repeat extraction. 

### 2.2. Data Extraction and Quality Assessment

Two investigators extracted data from each included study, which were then reviewed independently by a third investigator (ER). The extracted information included details of clinical and demographic characteristics (e.g., age, gender), number of patients, GERD symptoms, and endoscopic findings. All repetitive information was excluded.

## 3. Results

### Search Results

A total of 140 studies were identified using our search criteria for screening. After an assessment of the title, according to our inclusion criteria (endoscopic findings as a primary outcome), 125 articles were removed as they were not relevant and specific for the topic. Finally, 15 studies remained for content review. Of the 15 studies, 8 were identified as being not relevant for the paper, as they mixed aspects of our primary key questions with other aspects of their proper research. Thus, a total of 7 primary studies meeting the inclusion criteria, and one meta-analysis, were identified after careful screening, and are summarized in Table 1.

## 4. Discussion

There is a vast amount of controversy regarding the symptoms of GERD and their correlation with previous bariatric surgery, and the anatomical situation of the esophageal hiatus. Studies concerning the analysis of the initial anatomical situation of an obese patient, which occurs whenever a LRYGB or LSG is performed, are extremely limited. In fact, there is little evidence that allows us to promote any single technique according to the hiatus situation. It seems that when large hiatal hernias are present in an obese patient in the preoperative setting, LRYGB is the most reasonable surgery [20]. However, some authors have developed protocols to include large hiatoplasties with mesh placement, and even antireflux surgical strategies, while performing LSG.

As we can see in Table 1, sometimes there is a lack of clear correlation between self-reported reflux symptoms and endoscopic findings [21]; bariatric surgery caused either de novo GERD or the aggravation of existing GERD. Gu et al. [16] found the improvement or remission of GERD (40.4%) in LSG and (74.2%) in LRYGB patients. LRYGB had a better effect on GERD (OR = 0.19, 95% CI: 0.12–0.30, *p* < 0.001) compared to LSG, especially within 3 years and > 3 years [16]. 3534 patients were included, 1918 were subjected to LSG and 1616 to LRYGB, the appearance of new GERD was 9.3% and 2.3% after LSG and LRYGB, respectively (179 in LSG and 37 in LRYGB). In the global analysis, the authors found that LSG showed a higher risk of GERD than LRYGB (OR = 5.10, 95% CI 3.60–7.23, *p* < 0.001); the results were consistent in the subgroup analysis according to type of study, follow-up time, low heterogeneity, and the risk of GERD after surgery. 

LSG is known to have long-term complications related to GERD when compared to LRYGB [17]. The appearance of GERD and its evolution after LSG or LRYGB is related to the drawbacks of these two types of bariatric surgeries. The signs of improvement in GERD following LSG include: the acceleration of gastric emptying [22], a decrease in intra-abdominal pressure associated with the weight loss, and a decrease in acid production with the resection of the stomach [1]. However, the worsening of GERD following LSG includes predominantly the dissection of the gastroesophageal junction [1], the decrement of the lower esophageal sphincter pressure [23], and the modification of the anatomy at the angle of Hiss (responsible for the immediate postoperative GERD) [24].

Some authors defend a routine endoscopy prior to bariatric surgery. Moulla, Y. et al. [25] published a study where the incidence of upper-GI pathologies detected prior to bariatric surgery was evaluated. They reported a total of 636 obese patients with a median BMI (body mass index) reaching super-obesity (Mean BMI: 49 kg/m^2^ [range 31–92]). Endoscopy detected pathological conditions, such as Helicobacter pylori (Hp) gastritis, in more than 20% of endoscopies, and gastric or duodenal polyps in 6.8%. They also described peptic ulcers in 3.5% of the patients. Reflux esophagitis could be detected in 21.9%. Barrett’s esophagus (BE) was histologically diagnosed in 15.0%, while BE was suspected endoscopically in only 11.3%. Esophageal adenocarcinomas were detected in 0.5%. However, these endoscopic and pathological findings only changed the operative strategy in 1.6% (10 patients) [26]. In agreement with the other authors, they concluded that preoperative upper-GI endoscopy identifies a wide range of abnormal endoscopic findings in obese patients, which may have a significant impact on decision-making, so it should be considered in all obese patients before a bariatric procedure [25]. Unfortunately, the influence of preoperative gastroesophageal reflux disease cannot always be evaluated due to the lack of routine preoperative upper endoscopy, especially in LRYGB [18,26].

On the other hand, Saarinen T. et al. analyzed the clinically significant findings in preoperative endoscopy and how they associate with preexisting GERD symptoms and premalignant lesions. They included 1474 operations, of which 71.0% were RYGB and 27.6% were LSG. They reviewed 86.5% preoperative endoscopic reports, finding that 50.7% were normal and 23.0% had a clinically significant finding, relevant for LSG (HH, esophagitis, BE, esophageal dysplasia). In the same analysis, for patients undergoing LRYGB, only 1.6% of the patients had significant findings relevant for LRYGB (peptic ulcer, atrophic gastritis, gastrointestinal stromal tumor, GIST) and 3.2% of the patients with LSG were converted to LRYGB, due to GERD [27]. These data show that preoperative endoscopy is indicated before LSG, but could be avoided in asymptomatic patients for LRYGB without the risk of gastric pathology.

During the follow-up, upper gastrointestinal symptoms are often difficult to interpret after LSG and LRYGB [13]. Abdominal pain is an especially common symptom after bariatric surgery and endoscopic exploration is commonly used to discard any type of potential complication. However, most common complications can cause gastrointestinal symptoms, such as gallstone disease, marginal ulcer, internal herniation, and stomal stenosis in LRYGB. Stenotic sleeves can also lead to abdominal symptoms, including GERD, nausea, and vomiting [28,29,30]. Complementary explorations need to be performed, such as an upper gastrointestinal X-Ray and upper endoscopy. With these imaging tests, we aim to understand what could produce complications. There is a lack of evidence and clinical data to help us understand which patients will benefit more from an upper endoscopy, because of the high number of upper endoscopies without relevant findings. 

Tai et al. [3] show that GERD with morbid obesity and LRYGB substantially improves not only the reflux symptoms but also the erosive esophagitis. Csendes et al. [4] concluded that LRYGB is effective to control pathological gastroesophageal reflux in patients with morbid obesity and Biter et al. [2] demonstrated in a randomized controlled trial that patients who underwent LSG have significantly higher GerdQ scores at both 2 and 12 months, postoperatively, than patients who underwent LRYGB.

An abnormal upper gastrointestinal endoscopy series and dysphagia as a reason for GERD for referral were associated with stomal stenosis in LRYGB [9,31] and margin ulcer [30] after LRYGB. Marginal ulcer incidence has been described as ranging from 0.1% to 4.6% after LRYGB [32]. It is most prevalent in the first year after surgery, but not restricted to the first year, with a mean time between surgery and the first symptoms of approximately 4 months [33]. Stomal stenosis has reported a prevalence ranging from 3.1% to 7.8% [34]. Some of these patients might refer to GERD symptoms, however, it is infrequent to find clear esophagitis in LRYGB patients complaining of upper gastrointestinal symptoms [29,30].

In contrast to the situation following LSG, Gastroesophageal reflux after sleeve gastrectomy is mainly due to the progressive dilatation of cardia, or the presence of a hiatal hernia for an extended period after surgery. It is, however, very relevant and important to notice that asymptomatic patients might have very abnormal endoscopic findings, including esophagitis grades B or C, or even Barrett’s esophagus [35]. The lack of clinical correlation between endoscopy findings and patient symptoms, especially after LSG, have been recently described in the long-term follow-up after laparoscopic sleeve gastrectomy [36]. Genco et al. first demonstrated that almost 17.2% of 110 patients undergoing LSG will develop Barrett’s esophagus (BE) at a median follow-up superior to 55 months. Interestingly, the authors also reported that 26.4% of patients with Barrett’s had no GERD symptoms. This preliminary finding was also confirmed by Soricelli et al. reporting that 21% of patients with evidence of BE had no symptoms of GERD. [37] Felsenreich et al. [18] also reported a high prevalence of BE in 15% of the patients, 10 years after SG, associated with 45% of esophagitis, while only 37% of patients complained of GERD symptoms. From these data, we can extract that the need for systematic endoscopic control beyond 5 years after LSG is independent to the presence of GERD symptoms. More recently, the ASMBS position statement on the rationale for the performance of upper gastrointestinal endoscopy, before and after metabolic and bariatric surgery, stated that a clinical evaluation by symptoms alone does not reliably diagnose or rule out GERD, and that patients with conclusive and objective evidence of preoperative GERD are better served by current techniques of RYGB, rather than SG [38]. Both the expert panel and the current available literature recommend an endoscopy screening of all patients with gastrointestinal symptoms, including GERD symptoms. Therefore, it would be reasonable to perform an endoscopy on patients 3 years after SG, irrespective of GERD symptoms, to rule out Barrett’s esophagus [38].

Another debate is the correlation between symptoms and time after surgery. Shorter time between LRYGB or LSG and the onset of symptoms can lead to technical aspects, such as stenosis of the anastomosis or a kinking of the LSG, which causes an abnormal functioning of the sleeve, along with nausea, vomiting, and GERD [39].

In a recent meta-analysis, in which GERD symptoms were evaluated after bariatric surgery, LSG was converted to LRYGB for patients suffering from severe GERD [16]. The conversion rate was approximately 1.82–8.91% [16]. RYGB is an antireflux procedure because the new anatomy avoids bile reflux, and the small lesser curvature in the gastric pouch excludes the acid-secreting gastric fundus dramatically. It should be stressed that a conversional RYGB due to GERD from a SG should be done with a short gastric pouch.

The conversion from LSG to LRYGB to treat severe GERD after LSG (with or without relevant endoscopic findings) requires complementary studies to detect anatomical defects, such as hiatal hernia de novo [18,31,40,41]. As the effect of a RYGB conversion on the evolution of Barrett’s mucosa is still unclear, endoscopic surveillance should be wisely performed in this setting. Bariatric surgery can improve GERD symptoms; however, new data need to emerge in the next few years, especially to avoid the lack of correlation between endoscopy and symptoms.

## 5. Conclusions

Endoscopic findings after bariatric surgery are controversial and might have a bad correlation with clinical symptoms. It seems that GERD outcomes for patients with previous GERD or de novo symptoms, are better in patients after being operated by LRYGB, than by LSG. There is a need to study in more depth the meaning of these GERD events, both in the preoperative period and after any bariatric procedure. Multifactorial reasons appear to be related to GERD. In order to avoid severe complications in the mid-long term, such as the appearance of the worsening of GERD, we should recommend LRYGB conversions. Endoscopy should also be recommended after bariatric surgery when clinical symptoms are present and after 3 years of a SG, even without symptoms, in order to diagnose Barret’s esophagus.

## Figures and Tables

**Table 1 medicina-57-00506-t001:** RYGB-Roux en Y gastric bypass, SG-Sleeve Gastrectomy, Pts—patients; yrs—years; mo—months; Endoscopic findings were graded according to the Los Angeles Classification, Barrett’s esophagus was biopsy proven. NA: not applicable.

Study	Journal/Year	Number of Patients	Age (Years)	Gender (Female)	Interval Time from Surgery to Endoscopy	Endoscopic Findings (Pts) [%]	Weight Loss Results at Evaluation	Additional Comments
Borbély et al. [12]	SOARD 2018	47 (100%RYGB)	36.5 (19–67)	27 (57.4%)	3.8 yrs (3–12)	Esophagitis (C and D LA) (5) [10.6%] Barrett’s esophagus (7) [14.9%] Marginal ulcers (4) [8.5%]	30.3 (20.3–47.2)	Esophagitis improved in 19 patients (41.3%), remained similar in 14 (30.4%), and worsened in 13 (28.3%).
Boerlage et al. [13]	SOARD 2020	98 (100%RYGB)	41 (±10.0)	223 (89.2%)	7 Mo (2–16)	Reflux esophagitis (6) [2.4%] Marginal ulcer (46) [18.4%] Stomal stenosis (26) [10.4%] Bleeding (7) [2.8%] Candida esophagitis (3) [1.2%] Food impaction (2) [0.8%]	TWL% 25.7 (±12.9)	
Huang et al. [14]	Gastrointestinal endoscopy 2003	49 (100%RYGB)	46 y (28–65)	42 (85.7%)	49 Mo	Marginal ulcer (13) [27%] Stomal stenosis (9) [19%] Esophagitis (2) [0.4%]		
Signorini et al. [15]	Surgical endoscopy 2019	227 RYGB 80 (35.2%) SG 147 (64.8%)	44.9 (36–53)	179 (78.9%)	2 yrs	HH de novo SG group (20) [46%] RYGB group (1) [2%] HH+EE postoperative SG group (16) [11%] RYGB group (8) [10%]	NA	SG had more de novo EE than GBP (25% vs. 5%, *p* = 0.001). EE improved in 10%, was resolved in 31.2%, worsened in 2.5% and remained unchanged in 10% of RYGB cases.
Lihu Gu et al. [16]	Obesity Surgery 2019	23 studies ** LSG 2463 (49.2%) RGYB 2537 (51.3%)	NA	NA	NA	Correlation of endoscopic findings and GERD symptoms.	NA	LSG was associated with a higher risk of GERD than LRYGB (odds ratio [OR] = 5.10 [3.60–7.23], *p* < 0.001).
Dimbezel et al. [17]	Obesity Surgery 2020	48 100% LSG	49.63 (±11.69)	42 (87.5%)	62.4 mo	Esophagitis (A and B LA) (17) [35.4%] Esophagitis (C and D LA) (1) [0.2%] Barrett’s esophagus (4) [8.3%]	40 ± 1.89 kg/m^2^	RYGB conversion improved EE (14) [29.2%].
Felsenreich et al. [18]	Obeisty Sugery 2017	53 100% LSG	38.4 (±12.4)	42 (79%)	129 mo	Hiatal hernia de novo (9) [16.9%] Columnar lined esophagus (10) [5.3%] (symptomatic reflux, 7; no reflux, 3). Barrett’s esophagus (3) [15%]		CLE is significantly longer in patients who suffer from symptomatic reflux (4.0 mm) than in patients who do not (2.3 mm) (*p* = 0.013). RYGBP conversion due to reflux (8) [14%]
Braghetto et al. [19]	Arq Bras Cir Dig 2021	39 100% LSG	43.7 (±8.5)	34	5.6 (±2.5 yrs)	Erosive esophagitis (33) [84.6%] Esophagitis (A and B LA) (28) [71.7%] Esophagitis (C LA) (5) [12.8%] Barrett’s esophagus (5) [12.8%]	38.4 + 13.4 kg/m^2^	mean time of appearance of reflux symptoms after surgery was 26.8 + 24.1 mo.

** Systematic review and meta-analysis include 23 studies: 6 RCTs, 6 prospective observational studies, and 11 retrospective observational studies.

## Data Availability

All the data are available from the corresponding author upon reasonable request.

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
