# Peer review of "GERD after Bariatric Surgery. Can We Expect Endoscopic Findings?"

_medicina, 2021, doi:10.3390/medicina57050506_

Round 1

Reviewer 1 Report

  1. Many statements need references. Like when you state that "LSG seems to achieve equal weight losses LRYGB)
  2. Why if you found 140 records, you removed 125 based on Title and abstract. (Please expand)
  3. When you mentioned that  "finally 15 studies remained for content review. Of the 15 studies 8 were identifies."  For what? What happened. The System of elimination or selection is not clear

Author Response

Reviewer 1

Comments and Suggestions for Authors

  1. Many statements need references. Like when you state that "LSG seems to achieve equal weight losses LRYGB)

We have reviewed the manuscript and focusing on the sentences that would require a reference missing.

  1. Why if you found 140 records, you removed 125 based on Title and abstract. (Please expand)

Data review was done according to the process resumed in the article “A total of 140 records were identified by the initial search of the PubMed/MEDLINE databases. Of these, 125 papers were excluded after screening by title and abstract. Our initial analysis included a pre screen to identify the clearly irrelevant reports by title, abstract, and keywords of the publication. Two other independent reviewers (AC. and SS.) then assessed the studies for relevance, inclusion, and methodological quality. The studies were classified as relevant (meeting all specified inclusion criteria); possibly relevant (meeting some but not all inclusion criteria); and rejected (not relevant to our review). Two reviewers (ER. and RV.) independently reviewed the full-text versions of all studies classified as relevant or possibly relevant. Any disagreements were resolved by repeat extraction.”. We have modified the text in order to expand the explanation.

  1. When you mentioned that  "finally 15 studies remained for content review. Of the 15 studies 8 were identifies."  For what? What happened. The System of elimination or selection is not clear

We have included an explanation to clarify this methodological aspect. In fact, once reviewing the text, these articles included more information regarding another aspects of obese patients but not answering specifically to the question regarding the role of endoscopy or findings.

Reviewer 2 Report

I thank and congratulate the authors for this interesting review.
The topic addressed in the study, in recent years has become fundamental for evaluating the effects of bariatric surgery and the long-term results on the quality of life of the operated patients. For this reason, it is a pity that despite the importance of the topic, only 8 studies in the literature evaluate the relationship between clinical symptoms and endoscopic findins after bariatric surgery. Personally, I think that a postoperative endoscopic protocol should be adopted for all operated patients, independently of LSG or LRYGB, with endoscopies at 3, 5, and 10 years after surgery. But this remains my personal opinion.

As for the present study, the only real shortcoming remains the discussion that is too long with often redundant concepts associated with an English that is sometimes difficult to follow. This is a study that should be read extensively given the many insights it gives on future clinical studies. For this reason, the discussion should be reviewed by eliminating concepts that have been expressed too many times and trying to make a more linear text.

Sincerely

Author Response

We have reviewed the manuscript and avoid the redundant sentences during the discussion

Reviewer 3 Report

1. Please explain / detail  what exactly is meaning "true pathological GERD" in "... a high percentage of patients with severe GERD symptoms do not have true pathological GERD on objective testing".

2. Some statements are not supported by references : "It seems that when large hiatal hernias are present in the preoperative setting in an obese patient LRYGB seems the most reasonable surgery"

or "...whereas QOL did not differ significantly",

or "A prevalence ranging from 3.1% to 7.8% has been reported for stomal ..."

3. Please detail "Complementary explorations need to be performed ..." and explain more precisely "... diagnostic yield is relatively low" and "... a kinking of the LSG".

4. It can be a valuable information to detail "patient characteristics" which emerges from this statement: "... only few studies evaluated patient characteristics that are associated with relevant findings at upper endoscopy and mainly in LRYGB".

5. Proofreading recommended - eg: "who seekbariatric"; "It is then when endoscopic findings can be found that there is sometimes a lack of clear correlation between what expresses the patient and endoscopic findings ..."; "noy only the reflux..."; or "...any relevant endoscopic find me", or "there is not enough for me in place for you to wait lost anatomical changes after surgery...", or "They reported a tota of", etc.

6. I recommend a clearer systematization of the complications occurred after the different types of surgical technique used in the different studies, or appeared in relation to the moment of the surgery - perhaps for example presented in the graphical form of a table, etc.

7. I also recommend a more systematic presentation of the types of explorations used in the diagnosis of complications after bariatric surgery and their benefits.

Author Response

Reviewer 3

Comments and Suggestions for Authors

1.Please explain / detail what exactly is meaning "true pathological GERD" in "... a high percentage of patients with severe GERD symptoms do not have true pathological GERD on objective testing".

We have included an explanation to clarify this aspect

  1. Some statements are not supported by references:

"It seems that when large hiatal hernias are present in the preoperative setting in an obese patient LRYGB seems the most reasonable surgery"

or "...whereas QOL did not differ significantly",

or "A prevalence ranging from 3.1% to 7.8% has been reported for stomal ..."

We have reviewed the manuscript avoiding some sentences and focusing on the statements that would require a reference missing.

  1. Please detail "Complementary explorations need to be performed ..." and explain more precisely "... diagnostic yield is relatively low" and "... a kinking of the LSG".

We have reviewed the manuscript attending these recommendations

4.It can be a valuable information to detail "patient characteristics" which emerges from this statement: "... only few studies evaluated patient characteristics that are associated with relevant findings at upper endoscopy and mainly in LRYGB".

We have reviewed the manuscript avoiding the statements without supporting reference.

5.Proofreading recommended - eg: "who seekbariatric"; "It is then when endoscopic findings can be found that there is sometimes a lack of clear correlation between what expresses the patient and endoscopic findings ..."; "noy only the reflux..."; or "...any relevant endoscopic find me", or "there is not enough for me in place for you to wait lost anatomical changes after surgery...", or "They reported a tota of", etc.

We have reviewed the manuscript attending these recommendations

  1. I recommend a clearer systematization of the complications occurred after the different types of surgical technique used in the different studies, or appeared in relation to the moment of the surgery - perhaps for example presented in the graphical form of a table, etc.

Thank you for your comment, we have focused in the in the utility of the endoscopy before and after bariatric surgery. We have reviewed the manuscript, but we have avoided to include a graphical form.

  1. I also recommend a more systematic presentation of the types of explorations used in the diagnosis of complications after bariatric surgery and their benefits.

We have done an effort in the writing to include a more systematic presentation and order of data.